# T-Cell Engagers in Solid Cancers—Current Landscape and Future Directions

**DOI:** 10.3390/cancers15102824

**Published:** 2023-05-18

**Authors:** Mohamed Shanshal, Paolo F. Caimi, Alex A. Adjei, Wen Wee Ma

**Affiliations:** 1Department of Oncology, Mayo Clinic, Rochester, MN 55902, USA; 2Cleveland Clinic, Cleveland, OH 44195, USA

**Keywords:** bi-/trispecific antibodies, T-cell engagers, solid tumors

## Abstract

**Simple Summary:**

There are multiple strategies to target cancer cells, and among the rapidly evolving field is the use of bispecific antibodies and T-cell engagers in the treatment of cancers. These drugs work by recruiting and activating T-cells, a type of white blood cell, to recognize and attack cancer cells. These agents consist of two different antibody fragments: one that binds to a tumor antigen on cancer cells and another that binds to the CD3 receptor on T-cells. Once the T-cell engager binds to both the cancer cell and T-cell, it brings the T-cell into close proximity to the cancer cell, leading to the activation of T-cells and the release of cytokines and cytotoxic molecules that kill the cancer cell. T-cell engagers have shown promising results in the treatment of a variety of hematological malignancies. Research is ongoing to explore their use in the treatment of variety of solid cancers. Nevertheless, T-cell engagers can cause side effects like cytokine release syndrome and neurotoxicity. More research is ongoing to determine their long-term safety and effectiveness.

**Abstract:**

Monoclonal antibody treatment initially heralded an era of molecularly targeted therapy in oncology and is now widely applied in modulating anti-cancer immunity by targeting programmed cell receptors (PD-1, PD-L1), cytotoxic T-lymphocyte-associated protein 4 (CTLA-4) and, more recently, lymphocyte-activation gene 3 (LAG3). Chimeric antigen receptor T-cell therapy (CAR-T) recently proved to be a valid approach to inducing anti-cancer immunity by directly modifying the host’s immune cells. However, such cell-based therapy requires extensive resources such as leukapheresis, ex vivo modification and expansion of cytotoxic T-cells and current Good Manufacturing Practice (cGMP) laboratories and presents significant logistical challenges. Bi-/trispecific antibody technology is a novel pharmaceutical approach to facilitate the engagement of effector immune cells to potentially multiple cancer epitopes, e.g., the recently approved blinatumomab. This opens the opportunity to develop ‘off-the-shelf’ anti-cancer agents that achieve similar and/or complementary anti-cancer effects as those of modified immune cell therapy. The majority of bi-/trispecific antibodies target the tumor-associated antigens (TAA) located on the extracellular surface of cancer cells. The extracellular antigens represent just a small percentage of known TAAs and are often associated with higher toxicities because some of them are expressed on normal cells (off-target toxicity). In contrast, the targeting of intracellular TAAs such as mutant RAS and TP53 may lead to fewer off-target toxicities while still achieving the desired antitumor efficacy (on-target toxicity). Here, we provide a comprehensive review on the emerging field of bi-/tri-specific T-cell engagers and potential therapeutic opportunities.

## 1. Introduction

The use of monoclonal antibody (mAb) technology in anti-cancer therapy has evolved rapidly since the Food and Drug Administration first approved the use of the anti-CD20 mAb rituximab in 1997 for treating relapsed CD20-positive B-cell lymphoma. Improvements in gene sequencing, proteomics and computational platforms resulted in the production of antibodies with increasing affinity to antigenic epitopes and efficacy [1]. Targeting tumor-associated antigens (TAA) continues to be a major focus in anti-cancer drug development and has become particularly relevant in the era of immunotherapeutics because these antigens may be targeted to elicit antitumor immune responses [2,3]. However, this approach can be hindered by several resistance mechanisms, including but not limited to downregulation of TAA expression by cancer cells, activation of signaling pathway granting cancer cells resistance to apoptosis as well as tumor microenvironmental factors impeding mAb activity [4]. Furthermore, inducing immune responses that alter antibody-dependent cellular cytotoxicity and complement-dependent cytotoxicity may impair the efficacy of these mAbs [5].

Bispecific antibodies (BsAbs) are a proven anti-cancer drug platform capable of simultaneously targeting multiple TAAs and potentially overcoming these resistance mechanisms, and, in the era of immuno-oncology, they modulate/induce immune cell responses by simultaneously targeting the TAA(s) and antigens/receptors on the effector cells [6]. The majority of BsAbs are designed based on IgG molecules (Figure 1). The antigen-binding site (ABS) is an area between the heavy variable (V_H_) and light variable (V_L_) chains and is designed via the re-arrangement of the complementarity-determining regions (CDRs) [7].

Optimizing the CDRs is important for higher affinity (on-target efficacy) and avoidance of bystander cells (off-tumor toxicity) [8]. The design of an effective BsAbs involves multiple processes, and for the purposes of this review we will summarize it into four key steps: (A) lead identification, which is the detailed analysis/identification of the antibody-binding site for an antigen e.g., protein docking [9,10]; (B) predicting the antibody structure through proteomic computational methods, e.g., AlphaFold [11]; (C) lead optimization, which is the prediction of the binding affinity between the antibody (paratope) and the antigen (epitope), e.g., the PInet [12]; and finally (D): for the antibody to be effective, it does require good solubility to avoid overt immunogenicity. Lead identification, optimization and assessment of the relative positions of both parts of the paratope–epitope complex are crucial steps for the development of successful therapeutic antibodies [13].

BsAbs are designed in a way that one target is the neoepitope of cancer cells (TAA) and the other target site is dedicated to engaging with targets that can facilitate an antineoplastic effect. The antineoplastic activity can be achieved through several mechanisms: (1) direct engagement with surface antigens of immune effector cells, such as CD3 in T-cells, CD16 in NK cells or CD47 in macrophages; (2) engagement with receptors that modulate T-cell response; and (3) interaction with other signaling pathways.

The most common BsAbs are the bispecific T-cell engagers (BiTEs), which act through the simultaneous engagement of TAAs and CD3, resulting in the activation of T-cells irrespective of MHC, with the resultant release of perforins and granzymes [14]. In solid tumors, the success in targeting GP100 through construction of gp100/HLA*0201 fused to anti-CD3 single-chain variable fragments (scFv) led to tebentafusp’s approval for the treatment of uveal melanoma.

BsAbs’ T-cell-modulating targets include those that are immune-inhibitory such as PD-L1/CTLA-4 [15] and those that are immune-stimulatory such as the TNF receptors OX40, CD27 and CD137 (4-1BB) or the T-cell costimulatory receptor CD28. Engagement of one or more of these receptors may enhance the antitumor immune response or activate exhausted tumor-infiltrating lymphocytes in the tumor immune microenvironment [16].

Antibody–drug conjugates are another approach that utilize the mAb platform for targeted delivery of cytotoxic agents [17]. The use of trastuzumab deruxtecan in the treatment of HER2+ breast cancer is a successful example in which the BsAb binds to HER2/CD63 or HER2/PRLR and concurrently facilitates the internalization and lysosomal degradation essential for the action of the cytotoxic payload [18,19].

At the time of the writing of this review, there were a total of seven BsAbs approved for the treatment of different malignant diseases: amivantamab (EGFR/cMET), blinatumomab (CD3/CD19), catumaxomab (CD3/EpCAM), mosunetuzumab (CD3/CD20), tebentafusp (GP100/CD3), teclistamab (CD3/BCMA) and zenocutuzumab (HER2/HER3) [20,21].

Here, we focus on conditional T-cell engagers in solid tumors using the bi-/trispecific antibody platform and contrast this with immune effector cell therapies such as CAR-T therapy. We will also discuss the potential benefits of targeting intracellular TAAs and provide a summary of currently approved agents and those in development for the treatment of solid tumors.

## 2. Structural Mechanism of T-Cell Engagers

Bispecific antibodies (BsAbs) are divided into two major categories based on their structure and mechanism of action.

### 2.1. T-Cell Engagers without FC Fragment

Characterized by a lack of immune-mediated target cell killing (ADCC), as they lack the FC domain, these agents have lower stability and a short plasma half-life but have better tissue penetration given their lower molecular weight. These are particularly useful in targeting the central nervous system (CNS), and examples include nanobodies and svFC [22,23] (Figure 2A).

### 2.2. T-Cell Engagers with FC Fragment

These agents have the potential to exert actions related to their FC fragment, including ADCC. BsAbs with FC fragments are characterized by a longer half-life and higher stability. The design examples include the knob-in-hole technique in which the heavy chain is engineered with a knob, and the other heavy chain consists of a hole [24]. The disadvantages of including an FC fragment on a BsAb are both structural and functional. The structural disadvantage is that the large size of the molecule impedes tissue distribution and access to neoplastic cells. In addition, the non-covalent binding of the two variable domains via a hydrophobic interface is more challenging from the design and manufacturing aspects (Figure 2B). From a functional standpoint, the presence of an intact FC domain decreases T-cell trafficking and limits the antineoplastic activity, though it can be improved by FC silencing via FC fragment modifications [25].

## 3. T-Cell Engagers in Solid Tumors

Most approved BiTEs are used in the treatment of hematologic malignancies. While TAAs have been identified both in hematologic malignancies and solid tumors, most are also present in normal cellular counterparts, which results in “on-target” toxicities. These on-target toxicities in normal tissues are more manageable in hematologic malignancies than in solid tumors. For example, CD19 targeting results in B-cell aplasia and hypogammaglobulinemia and increases the risk of infection; such infections can be managed by being vigilant clinically and the prompt use of antibiotic(s). On the other hand, targeting epidermal growth factor (EGFR) in lung cancer leads to generalized cutaneous toxicity and cardiotoxicity in the treatment of HER2-positive breast cancer using trastuzumab [26,27].

These “on-target” toxicities can further be mitigated through the use of conditional T-cell engagers (cTCE), which are inactive precursors of TCEs activated by tumor-associated proteases. This induces TCEs to kill tumor cells expressing the target antigen in the tumor microenvironment without affecting distant tissues. The decreased toxicity of cTCEs potentially confers upon them a superior therapeutic index and may allow the administration of a higher dose. An example is ProTriTACs, where a half-life-extending albumin-binding domain masks the CD3-binding domain [28,29].

Intracellular TAA targeting may sidestep the challenges of mAb targeting of shared cell surface TAAs between tumor and normal tissues in solid tumors. This strategy may lead to higher tumor cell killing (on-target/on-tumor effect), decreased risk of damaging normal tissues (on-target/off-tumor toxicity) as well as lower bystander toxicity (off-target/off-tumor toxicity).

However, intracellular TAA targeting is currently challenged by a lack of effective tools to penetrate the cell surface and deliver the antibody arm of TCEs into neoplastic cells. This is an area of active research, and novel approaches include technologies that promote the expression of intracellular TAAs or enhance the delivery of BsAbs into tumor cells.

### 3.1. Targeting Tumor-Associated Peptides Presented by MHC

Aberrant intracellular proteins can be presented by MHC class 1 molecules on the cell surface (Figure 3E). One example is p53: a small fraction of intracellular p53 is degraded via proteosomes and can be presented by HLA on the cell surface. Specific peptides of mutated p53 (R175H) can bind to HLA-A*02:01 on the surface of cancer cells. Hsiue et al. developed CD3 engagers that bind specifically to the p53-R175H peptide–HLA complex with high affinity and resulted in tumor cell killing [30,31,32].

A similar strategy is the use of oncolytic vaccines such as MAGE-1 and NY-ESO-1. These are incorporated into the tumor genome and are translated into antigens which, when degraded, are presented by MHC to the extracellular surface of tumor cells, making them recognizable by T-cell engagers. An example is melanoma-associated antigen A4 (MAGE-A4) solid cancers in patients with the HLA-A*02:01 genotype. MAGE-A4 is processed intracellularly, resulting in peptide fragments that are co-presented with HLA-A*02:01 (Figure 3E). Afamitresgene autoleucel is an HLA-restricted autologous T-cell therapy that targets MAGE-A4 that was evaluated in a phase 1 trial in solid tumor patients with HLA*02:01. The overall response rate was 24% (all partial response). All patients experienced grade 3 and above hematologic toxicity; 55% of patients experienced =< grade 2 cytokine release syndrome [33].

### 3.2. Intracellular Delivery of Antibodies

The delivery of larger antibodies to intracellular targets is an area of intense research, and approaches under investigation include the use of liposomes, cell-penetrating peptides, nanoparticles, cytosol-penetrating antibodies, TCR-Like antibody and the use of viral vectors (Figure 3A–F). The successful targeting of intracellular p53 and KRAS mutants had been demonstrated in preclinical studies [34].

### 3.3. Successful Examples of T-Cell Engagers in Solid Tumors

This section highlights the importance of T-cell engagers in the treatment of advanced solid tumors. Some of these example are FDA-approved, while others are in later stages of drug development. (Figure 4, summarize therapeutic targets in solid tumors).

GP100

Melanoma-associated antigen (gp100) is a membrane-bound protein expressed on the surface of melanocytes and most malignant melanoma. T-lymphocytes recognize gp100 presented by HLA proteins such as HLA-A*02:01. Vaccination with gp100 peptide vaccine was evaluated in a phase II randomized trial in the treatment of metastatic melanoma in combination with IL2; the overall response rate (ORR) was 16%, and overall survival (OS) was 17 months [35]. This suggested gp100 as a potential TAA, which led to designer BiTEs’ with a high-affinity TCR-binding domain and an anti-CD3 T-cell-engaging domain. Such a construct facilitates and redirects T-cells to attack gp100-expressing tumor cells. Tebentafusp is a gp100/CD3 BsAb and was evaluated in a phase 1 study of 42 patients with metastatic uveal melanoma. The study found that tebentafusp was generally well-tolerated, with the most common adverse events (AEs) being fever (91%), rash and pruritis (83%), fatigue (71%) and chills (69%). The overall response rate was 11.9% (95% CI, 4.0 to 25.6). The median overall survival was 25.5 months (range, 0.89–31.1 months), and the 1-year overall survival rate was 67% [36]. The subsequent phase 3 trial in HLA-A*02:01-expressing patients with treatment-naive metastatic uveal and cutaneous melanoma compared tebentafusp against the investigator’s choice of treatment (pembrolizumab, ipilimumab or dacarbazine). At a median follow-up of 14 months, the tebentafusp-containing arm achieved superior survival compared to the investigator’s choice (median OS 22 months versus 16 months; 6-month PFS 31% versus 19%, respectively). A 1-year overall survival rate of 65% was reported for both patient cohorts [37,38]. This pivotal trial led to Food and Drug Administration (FDA) approval of tebentafusp in April 2022. Cytokine release syndrome (CRS) occurred in 89% of patients, though the adverse events improved during subsequent dosing. Only 2% of patients discontinued treatment due to treatment-related side effects, and there were no treatment-related deaths.

EPCAM

Catumaxomab is a T-cell engager with specificity for CD3 and epithelial cell adhesion molecule (EpCAM). Interestingly, this agent can be considered “trifunctional” because the mAb has a functional FC receptor capable of engaging FCγ receptors that induce immune reaction [39]. Catumaxomab was approved by the European Medical Agency in 2009 for treatment of EpCAM-positive carcinomas and malignant ascites. However, catumaxomab was withdrawn from the market in 2017 due to unacceptable CRS and high-grade liver toxicity which was fatal in some patients [40]. The severity of the toxicities was likely due to the high immunogenicity and wide expression of EpCAM in normal tissues such as the Kupffer cells of the liver.

PSMA

Pasotuxizumab, a CD3/PSMA T-cell engager, was evaluated in a phase 1 trial of patients with metastatic castrate-resistant prostate cancer (mCRPC). The BsAb was administered as a continuous IV infusion over 12 weeks. Of the 16 patients treated, 13 (81%) experienced grade ≥ 3 adverse events (AEs). The most frequent all-grade AEs were flu-like illness and fatigue, whereas the most frequent grade ≥ 3 AEs were decreased lymphocytes and infections (44%). Three patients had a reduction of >50% in serum PSA levels, two of them had long-term responses, and one achieved a near CR as assessed by PSMA PET imaging [41]. The development of pasotuxizumab was halted after this study in favor of acapatamab.

Acapatamab is T-cell engager targeting CD3/PSMA but has a longer half-life than pasotuxizumab. Acapatamab was evaluated alone or in combination with pembrolizumab in a phase 1 trial involving patients with mCRPC. In the monotherapy cohort, a ‘confirmed ≥ 30% PSA reduction’ was achieved in 27.6% of patients, a partial response in 20% and stable disease in 53.3%. However, 60.5% had grade 2 CRS, and 25.6% had grade 3 CRS. [42,43]. The CD3/PSMA T-cell engager is being compared to enzalutamide and abiraterone in patients with mCRPC in a phase 2 trial (NCT04631601). The study has completed accrual, and the results are pending.

CD33 (MDSC-targeting)

CD33 is a transmembrane receptor expressed on the surface of various myeloid cells, including myeloid-derived suppressor cells (MDSCs), and is known to promote tumor growth and suppress antitumor immune response in solid tumors [44]. AMV564 is a novel CD3/CD33 T-cell engager which induces antitumor immune response via T-cell-directed lysis of CD33 cells [45]. The agent was evaluated in a phase 1 trial which included 30 patients with advanced solid tumors. AMV564 was given via subcutaneous injections on days 1–5 and 8–12 of a 21-day cycle, either alone (20 patients) or in combination with pembrolizumab (10 patients) at a dose of 200 mg IV q3w. The monotherapy cohort dosing was 15, 50 and 75 mcg/day, while the combination therapy cohort dosing was 5, 15 and 50 mcg/day. Complete response was achieved in patients with ovarian cancer in the monotherapy cohort. The agent was well tolerated, with 2 cases of grade 2 CRS at 75 mcg/day and no dose-limiting toxicities. The most common side effects were injection site reactions, fever, fatigue, anemia, hypotension and nausea. Subcutaneous injection of AMV564 resulted in relevant plasma exposure [46].

### 3.4. Toxicity Profile and Management of Toxicity of T-Cell Engagers in Solid Tumors

The major advantages of T-cell-engaging immunotherapeutics such as BsAbs and T-cell engagers over cellular therapeutics, e.g., CAR-T, include their “off-the-shelf” availability and their requiring no lymphodepletion prior to administration. Both treatment modalities (T-cell engagers and CAR-T-cells) have similar toxicity profiles, including CRS and immune effector-associated neurotoxicity syndrome (ICANS). These toxicities are less severe or less frequent in T-cell engagers and may be related to their dosing (Table 1 and Table 3) [47,48]. While CAR-T-cell treatment is often administered in a single dose, T-cell engagers may be administered repeatedly. As such, T-cell engager administration includes a period of step-up dosing to mitigate the risk of these toxicities and is followed by the full doses about 1 to 2 weeks after.

Cytokine release syndrome

The immune-mediated toxicities associated with treatment using CAR-T-cells and T-cell engagers are significant challenges requiring close attention. The binding of the target antigen induces T-cell expansion and the release of proinflammatory cytokines, which then leads to the recruitment of other endogenous immune cells such as macrophages and further escalation of the immune response [49]. The overactive and uncontrolled immune response results in CRS, which is characterized by fever and multiple-end-organ dysfunction/damage. This systemic toxicity is experienced by approximately 15% of patients treated using blinatumomab. The incidence is dependent on the BiTEs’ structure, affinity for CD3 engagement and TAA abundance, e.g., grade 3 and above CRS being reported in 2% of patients with MUC17/CD3 (AMG199) and 25% of patients with PSMA/CD3 (acapatamab) [42].

The risk can be mitigated by close monitoring and premedication with corticosteroids. Step-up dosing can also reduce CRS risk by starting at a lower dose and escalating during the first cycle of treatment. The use of a subcutaneous route has been proposed to decrease the risk of CRS [50]. A recent computational study showed that the co-administration of the interleukin (IL)-6 receptor-blocking mAb tocilizumab and/or anti-TNFα may reduce the incidence without compromising antitumor activity [51].

The management of low-grade CRS includes the use of antihistamines, antipyretics and intravenous fluids. CRS of grade 2 and above requires admission to the hospital and may require pressors for hypotension, as well as supplemental oxygen for respiratory distress [52,53]. Severe CRS will require intensive care unit care with early initiation of high-dose corticosteroids—for example, methylprednisone at 1 mg/kg/day and tocilizumab at 8 to 12 mg/kg based on body weight. Tocilizumab can be repeated after an interval of 8 h, and it should not exceed 4 doses in total [54]. Supportive therapies include the use of intravenous vasopressors or mechanical ventilation to assist with respiratory distress.

Neurotoxicity

Immune effector-associated neurotoxicity syndrome (ICANS) with T-cell engager treatment may manifest as mild confusion, headache, dysgraphia and, in more severe cases, encephalopathy. The pathophysiology of ICANS is not yet fully understood. Cerebral microvasculature damage from activated immune cells, endothelial damage, blood–brain barrier disruption and direct effects of cytokines have been hypothesized as underlying mechanisms [55].

The neurotoxicity risk from T-cell engagers may be mitigated by step-up dosing and pre-treatment with corticosteroids. Once it has occurred, management consists primarily of aggressive supportive care and corticosteroids. Tocilizumab, however, is not recommended for neurotoxicity, as it can increase IL-6 plasma concentration and exacerbate ICANS.

The management of ICANS often involves supportive care measures such as hydration, electrolyte balancing, the use of steroids and seizure management. For patients with more severe symptoms, more aggressive interventions may be indicated including mechanical ventilation or intracranial pressure monitoring. A neurologist or other specialist with experience in managing neurological complications of immunotherapy should be consulted if one has not been already.

The prophylactic use of corticosteroids has been shown to be effective in ameliorating the side effects of CRS and ICANS. The ZUMA-1 study showed the superiority of prophylactic use of dexamethasone at 10 mg for 3 days before axicabtagene ciloleucel (no grade 3 CRS) compared to the 13% incidence of CRS among patients who received steroids after an infusion of axicabtagene ciloleucel. The efficacy was not compromised in this approach, though there was a higher rate of ICANS [56].

Based on the above information, clinicians should be aware of the potential side effects of CRS and ICANS when using T-cell engagers. Early identification and prompt management are key to preventing the development of severe side effects. A multidisciplinary team with experience in immunotoxicity provides the best outcome.

TAA-specific toxicities

Other less common but important side effects include hepatoxicity and cardiac toxicity, which are mostly managed with supportive care and interventions described above. Table 1 lists toxicities described for solid tumor T-cell engagers.

T-cell engagers undergoing clinical evaluation in solid tumors

Publicly available databases including PubMed, meeting abstracts of major scientific meetings and ClinicalTrials.gov were searched for T-cell engagers that are in the early phases clinical testing (summarized in Table 2).

## 4. Conclusions

Bispecific T-cell engagers have emerged as a valid and clinically relevant anti-cancer therapeutic. The practical advantages of T-cell engagers over immune effector cell therapies, such as off-the-shelf availability and avoidance of complex cell handling facilities, broaden the applications of this class of immune therapeutics in cancer care (Table 3).

The majority of bispecific antibodies can be given on an outpatient basis; however, this can vary based on the drug toxicity profile and the clinical condition of the patient. Therefore, there are no standards of care of management, and many clinicians prefer to admit patients to inpatient care for cycle 1 to monitor for side effects before continuing via outpatient care.

Significant efforts are now underway in search of clinically relevant TAAs, optimizing molecular structures and identifying their roles in the treatment of hematologic and solid cancers. The majority of BiTEs in clinical development are directed towards extracellular proteins that represent 10% of the known targetable proteome [65]. Intracellular TAAs are the next frontier for T-cell engagers, and research to direct CD3 cells towards targets such as p53^R175H^ and RAS proteins on MHC molecules of tumor cells are underway [32,66]. Conditional activation of T-cell engagers is another approach to enhance tumor selectivity and on-target/on-tumor efficacy while lessening off-tumor toxicity. Here, inactive T-cell engagers become activated upon entering a tumor by either leveraging tumor proteases or modulating the pH of the microenvironment [67]. In summary, this class of immune therapeutics holds great promise, especially in highly heterogeneous solid tumors, and commands the attention of anti-cancer drug developers.

## Figures and Tables

**Figure 1 cancers-15-02824-f001:**
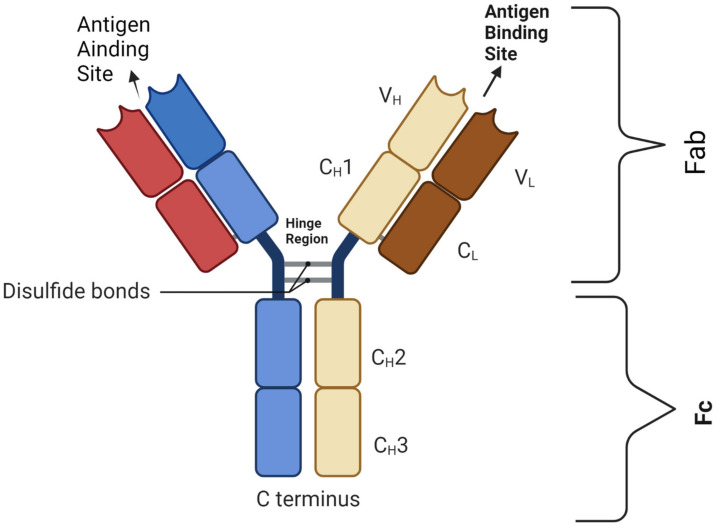
Basic BsAb structure showing the antigen-binding site that harbors the complementarity-determining region (CDR) segment, disulfide bond, light chains (L) and heavy chains (H); C: constant domain; V: variable domain, Fab: fragment antigen-binding domain; Fc: fragment-crystallizable domain. Figure created by BioRender.com (accessed on 14 May 2023).

**Figure 2 cancers-15-02824-f002:**
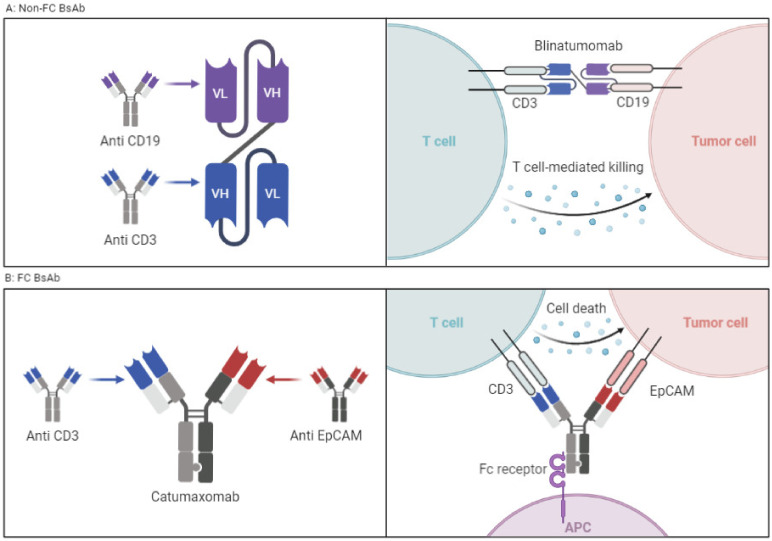
(**A**) BsAbs without FC portion connected via a linker. Example is blinatumomab. (**B**) Classic BsAbs with FC portion that requires antigen-presenting cells and activation of which results in ADCC. Example is catumaxomab. Figure created by BioRender.com (accessed on 14 May 2023).

**Figure 3 cancers-15-02824-f003:**
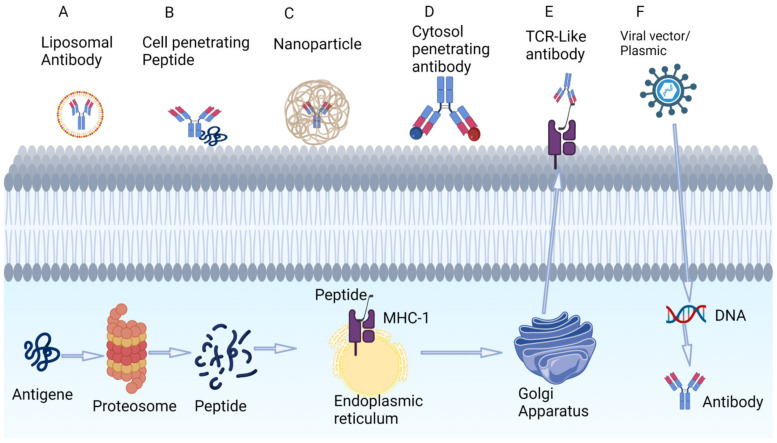
Techniques under investigation for targeting intracellular tumor antigens using antibodies. (**A**) Liposomal coating of antibodies. (**B**) Cell-penetrating peptides that adhere to the phospholipid bilayers. (**C**) Nanoparticles. (**D**) Cytosol-penetrating antibodies. (**E**) TCR-like antibodies that can target low-concentration intracellular peptide fragments that are presented to extracellular space via MHC. (**F**) Viral vectors that incorporate into the genome and produce intracellular antibodies. Figure created by BioRender.com (accessed on 14 May 2023).

**Figure 4 cancers-15-02824-f004:**
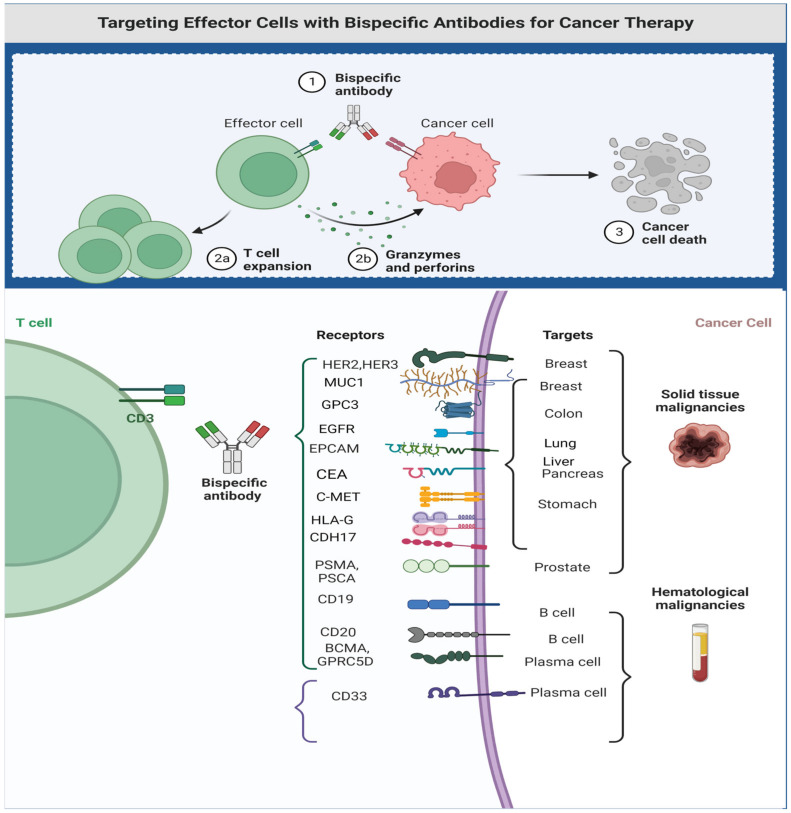
Therapeutic targets for bi- and trispecific antibodies. Figure created by BioRender.com (accessed on 14 May 2023).

**Table 1 cancers-15-02824-t001:** Unique toxicities based on therapeutic targets of T-cell engagers.

Target Antigen	Specific Toxicity
EPCAM	Immune-mediated hepatotoxicity
HER2	Hypotension, hypertension, tachycardia
PSMA	Hepatotoxicity
DLL3	Pneumonitis
Myeloid-derived suppressor cells (MDSC)	Anemia, hypotension, pruritis
EGFRvIII	Dermatologic toxicities, SJS, TEN
CEA	Hepatotoxicity

**Table 2 cancers-15-02824-t002:** Summary of T-cell engagers in development against solid tumors.

	Target Population	Phase of Study/References	Number of Patients	Results	Clinical Trial Number
CD3/HER2	Advanced HER2-positive breast cancer	Phase 2 [57]	32 patients, 8 patients had stable disease.	The median OS was 13.1, 15.2 and 12.3 months for the entire group, HER2-HR+ and TNBC patients, respectively. Plan for phase 3.	NCT03272334
EGFRvIII/CD3 (AMG596), (CX-904) EGFR/CD3.	EGFRvIII-positive GBM or malignant glioma	Phase 1/1b [58]	Total 14 patients.	1 partial response, 2 stable disease.	NCT03296696
	Tumor expression EGFR	Early phase 1	Plan for 100 patients	Not published	NCT05387265
Tyrosinase Related Protein 1 (TYRP1) (RO7293583)	Melanoma	Phase 1	20 patients	Not published	NCT04551352
MUC17/CD3	Advanced gastric, GE junction, CRC and pancreatic (AMG199)	Phase 1 [59]	Total 64 patients.	13 had PR, 17 SD. CRS > grade 3 occurred in 2%	NCT04117958
	Advanced liver cancer	Phase 2 [60]	11 Patients	Median PFS 4 months, median OS 13.2 months; 5 discontinued treatments due to severe side effects	NCT03146637
DLL3/CD3	Small cell lung cancer	Phase 1 [61]	Confirmed partial responses in 20% of patients and duration of response of 8.7 months	Phase 2 ongoing	NCT05060016
CEA (MEDI-565)	Advanced GI cancers	Phase 1	Total 39 patients,	11 patients have stable disease as best response.	NCT01284231
PSMA	Prostate cancer	Phase ½	LAVA-1207	Not published	NCT05369000
		Phase 1 [62]	AMG 509	Not published	NCT04221542
		Phase 1	BAY 2010112	47 patients, 12 patients had >50% decrease in PSA	NCT01723475
EpCAM	Advanced solid tumors	Solitomab: Phase 1 Catumoximab: Phase 2 [63]	Catumoximab and Solitomab: for malignant ascites both associated with sever toxicities precluding development of Solitomab.	Solitomab DLT in phase-limiting escalation. Catumoximab, withdrawn from market due to toxicities.	NCT00635596 NCT00836654
Myeloid-derived suppressor cells (MDSC)	Advanced solid tumors. with and without pembrolizumab	Phase 1	20 patients in monotherapy arm. 10 in combination arm.	Not fully published, study mentioned One CR. Study is going to phase 2.	NCT04128423
CLDN18.2 (AMG 910)	Gastric and gastroesophageal junction (G/GEJ) adenocarcinoma	Phase 1 [64]	Plan recruitment 34 patients	Not finished	(NCT04260191)
HLA-G	Advanced solid tumors	Phase 1	Actively recruiting	Not finished	NCT04991740

**Table 3 cancers-15-02824-t003:** Comparison of T-cell engagers and CAR-T therapies.

	T-Cell Engagers	CAR-T
Basic Structure	Bispecific antibody that binds TAA and CD3 on T-cells	Engineered T-cell that express engineered scFV fused to linker and activation domain
Source of T-cells	Endogenous T-cell activation	Requires ex vivo expansion of engineered T-cells
Availability	Outpatient	Inpatient and only at high-volume medical centers
Drug properties	Off the shelf	Must be engineered (2–4 weeks)
Dosing	Requires multiple doses, sometimes requires pump	One dose, sometimes multiple dosing if HLA is eliminated
Toxicity	Less CRS	Higher CRS and neurotoxicity
Lymphodepletion prior to treatment	Not required	Required
Operational Cost	High (USD 90,000) per course	Very high (USD 450,000 to 750,000)

CAR-T, chimeric antigen receptor-T-cells; CRS, cytokine release syndrome; scFV, single-chain, fragment variable antibody; TAA, tumor-associated antigen.

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
