# Peer review of "T-Cell Engagers in Solid Cancers—Current Landscape and Future Directions"

_cancers, 2023, doi:10.3390/cancers15102824_

Round 1

Reviewer 1 Report

This review article give a very good summary of the experience to date with clinical trials and approvals of bi-specific antibodies (BsAb) and their derivatives. It also provides a good discussion on structural and design principles for BsAbs and their comparison to CAR-T, for example, especially potential advantages of BsAbs. This review could be very useful in the field, although at this time some critique is provided and suggestions for improvement.

1) Most of the text uses the term off-target when what is more accurately meant is on-target, off-tumor. There is a section of the manuscript that uses the latter, more correct terminology. All the manuscript should be edited for this.

2) The title is more broad than the manuscript's material, and the manuscript at times includes tri-specific Abs in its scope, but there is no content on that. This manuscript reviews BsAbs and derivatives. The title and scope of manuscript should be edited to reflect this.

3) Figure 1 is not evidently a model of BsAb as it suggests, but appears to be a mono-specific, classic Ab.

4) The manuscript would be improved if a figure were provided displaying the principle of cTCE.

5) The following sentence should be edited to remove errors and improve clarity:  "...use of oncolytic vaccines such as MAGE-1 and NY-ESO-1. These are incorporated into the tumor genome and are translated into antigens ...".

6) Fig. 3E, it is unclear what is meant by having Golgi Apparatus arrow to surface peptide-MHC complex.

7) At the end of section 1, "... a total of 7 BsAbs approved for the treatment of different malignanat diseases". The attitude of this seems more positive than deserved, since although in history there were clinical trials, not all these were approved as drugs and there were even removals from the trial pipeline due to excessive toxicity. The authors' review of these 7 is appreciated and this information is good, here, but more must be said here to clarify that these 7 are not "approved" for clinical use.

8) "better penetration given their lower molecular weight". This is a claim that would benefit having peer-reviewed references specifically support this point. None of the Ab fragments approach organic drug-like size for tissue penetration. Does a 20 kDa BiTE really penetrate tissues better than 150 kDa classic Ab? They are all way too big compared with 200 dalton organic drug when measured by the 5-7 classic pharmacologic standards for drug development.

9) This reviewer likes very much the examples in section 3.3. It is suggested that the authors consider if they could provide a figure illustrating (even in simple icon form) each of these examples. Then the readers can quickly see which ones are BiTEs, which have Fc domains, etc.

10) "... toxicities are less severe or frequent in T-cell engagers and may be related to their dosing (table 2)". But Table 2 does not have information to make this point. The discussion of dosing with BsAbs is appreciated and it would help to see supporting information about that.

11) "... BiTES in clincial development are directed towards extracellular proteins that represent 10% of the known targetable proteome...". Please edit to improve clarity. BiTES are not directed against 10% of the known targetable proteome.

Despite the numerous points of critique, this is a good, useful review article. With improvements to clarity and focus, there can be high enthusiasm for a final manuscript.

If edits are made to clarify points above, language quality is sufficient.

Author Response

Dear

Thank you for your reviewing our review article. below are answers to your concerns.

1) Most of the text uses the term off-target when what is more accurately meant is on-target, off-tumor. There is a section of the manuscript that uses the latter, more correct terminology. All the manuscript should be edited for this.

Thanks for noticing that, i edited the manuscript and changed it to from off target to off-tumor.

2) The title is more broad than the manuscript's material, and the manuscript at times includes tri-specific Abs in its scope, but there is no content on that. This manuscript reviews BsAbs and derivatives. The title and scope of manuscript should be edited to reflect this.

Title changed to 

Bispecific antibodies in solid cancers– current landscape and future directions

3) Figure 1 is not evidently a model of BsAb as it suggests, but appears to be a mono-specific, classic Ab.

Changed the color coding to reflect the bispecific nature of the antibody.

4) The manuscript would be improved if a figure were provided displaying the principle of cTCE.

Figure added to abstract. 

5) The following sentence should be edited to remove errors and improve clarity:  "...use of oncolytic vaccines such as MAGE-1 and NY-ESO-1. These are incorporated into the tumor genome and are translated into antigens ...".

Modified, thanks

6) Fig. 3E, it is unclear what is meant by having Golgi Apparatus arrow to surface peptide-MHC complex.

In the endoplasmic reticulum, the peptides bind to MHC molecules, which are then transported to the cell surface via the Golgi apparatus and the secretory pathway. The resulting peptide-MHC complex is then displayed on the cell surface for recognition by T cells.

Added couple lines to the figure.

7) At the end of section 1, "... a total of 7 BsAbs approved for the treatment of different malignanat diseases". The attitude of this seems more positive than deserved, since although in history there were clinical trials, not all these were approved as drugs and there were even removals from the trial pipeline due to excessive toxicity. The authors' review of these 7 is appreciated and this information is good, here, but more must be said here to clarify that these 7 are not "approved" for clinical use.

Despite multiple trials, only few BsAb received FDA approval or progressed through advanced stages of drug development for the treatment of different malignant diseases e.g., amivantamab (EGFR/cMET), blinatumomab (CD3/CD19), mosunetuzumab (CD3/CD20), tebentafusp (GP100/CD3), teclistamab (CD3/BCMA) and zenocutuzumab (HER2/HER3). catumaxomab (CD3/EpCAM) was approved for treatment of malignant ascites and then withdrawn due to significant toxicity [20,21].

8) "better penetration given their lower molecular weight". This is a claim that would benefit having peer-reviewed references specifically support this point. None of the Ab fragments approach organic drug-like size for tissue penetration. Does a 20 kDa BiTE really penetrate tissues better than 150 kDa classic Ab? They are all way too big compared with 200 dalton organic drug when measured by the 5-7 classic pharmacologic standards for drug development.

BsAbs are particularly useful in targeting central nervous system (CNS) when delivered through dual-targeting liposomes or receptor-mediated transcytosis sources below: 

.        Kariolis, M.S., et al., Brain delivery of therapeutic proteins using an Fc fragment blood-brain barrier transport vehicle in mice and monkeys. Sci Transl Med, 2020. 12(545).

  1. Yin, W., et al., BBB-penetrating codelivery liposomes treat brain metastasis of non-small cell lung cancer with EGFR(T790M) mutation. Theranostics, 2020. 10(14): p. 6122-6135.

I can also remove the CNS and replace with just better tissue penetration? 

9) This reviewer likes very much the examples in section 3.3. It is suggested that the authors consider if they could provide a figure illustrating (even in simple icon form) each of these examples. Then the readers can quickly see which ones are BiTEs, which have Fc domains, etc.

Not sure if i can do that as each example will need a separate figure. But figure 4 list all potential targets. 

10) "... toxicities are less severe or frequent in T-cell engagers and may be related to their dosing (table 2)". But Table 2 does not have information to make this point. The discussion of dosing with BsAbs is appreciated and it would help to see supporting information about that.

Sorry, this should be table 3 and not 2 ( i made the correction in the note)

Added reference 48 To support dosing of BsAbs

11) "... BiTES in clincial development are directed towards extracellular proteins that represent 10% of the known targetable proteome...". Please edit to improve clarity. BiTES are not directed against 10% of the known targetable proteome.

Edited 

The majority of BiTEs in clinical development are directed towards extracellular proteins which represent only 10% of the known targetable proteome.

Reviewer 2 Report

This review aims to give an overview about bispecific T cell engagers for solid tumors. There is already various reviews available for this topic, thus this review should provide a differentiated view of the topic. In order to achieve this major modifications are required, currently, the review refers a lot to BiTEs developed in hematology sich as blinatumomab and/or terminated EpCAM bispecifics such as catumaxomab or EpCAM BiTE. I would expect a more up to date review focusing on the view success storie such as DLL3 and going more into depth e.g. showing how currently developed T cell engager formats look like. The list of T cell engagers is not up to date and should be reviewed, several molecules are missing or the status is not correct. Pieces about neurotoxicity and CAR-T cells should be removed when they refer to hematology, and be updated with solid tumor findings

Author Response

I disagree with this critic. if you read carefully, you will see list of all current and future BITE in solid tumors including DLL3. Please look to the tables created and read carefully through the article. 

If you look to published review articles related to BITES, it will also mention some extrapolation of data from hematology as we don't have much data in solid tumors yet. 

The purpose of this review is to update the readers with the direction of the field, and not how the format of Tcell engagers. target audience here are clinicians.

Mohamed Shanshal

Round 2

Reviewer 1 Report

Authors were positively responsive to review and have improved the paper.

Author Response

Appreciate your review of the article. Your previous points were valuable and helped add more to our review.

Based on your most recent reply (Authors were positively responsive to review and have improved the paper), seems there are no additional corrections. Please let us know if we missed something.

Thanks 

Mohamed Shanshal, MD

Reviewer 2 Report

as pointed out in my first review major revisions would be required, the authors only performed minor updated

Author Response

We appreciate Reviewer 2's review and comments.

Our goal of this manuscript is to provide an overview of the BITE's entering clinical testing in advanced solid tumors. As the reviewer pointed out, there is currently a lot of activity in hematologic malignancies and the development of this class is emerging in solid tumor clinically. As such, we aim to provide the principles and tools for solid tumor oncologists to understand this emerging field. To accomplish this, we gathered the information from Publicly available database including PubMed, clinicaltrials.gov and presentation from scientific meetings that are tested in solid tumors. We do not include agents only in preclinical evaluation and from confidential/privileged information. We thought that the references and agents relevant to solid tumors are up to date. 

Whilst we appreciate Reviewer 2's comments, we would appreciate specific examples to help improve our manuscript. otherwise, we find the comments to be vague and not actionable. We adamantly disagree with Reviewer 2's comments that a major rewrite is needed. 

Respectfully yours,